# A scoping review of community knowledge in malaria prevention and control programmes

**Faizul Akmal Abdul Rahim** *, **Mohd Amierul Fikri Mahmud,
Mohd Hatta Abdul Mutalip, Norzawati Yoep, Mohd Amiru Hariz Aminuddin,
Ahmad Mohiddin Mohd Ngesom**

Centre for Communicable Diseases Epidemiology Research, Institutes for Public Health, National Institutes of Health, Ministry of Health, Shah Alam, Malaysia

* faizul.fabregas@gmail.com

## Abstract

### Background

Malaria remains a significant global health challenge, particularly in sub-Saharan Africa and Southeast Asia. Despite considerable progress through biomedical interventions, persistent transmission underscores the need to examine additional influencing factors. This scoping review maps existing evidence on community knowledge in malaria prevention and control strategies, aiming to inform more targeted and culturally adapted interventions.

### Methods

Adhering to the Preferred Reporting Items for Systematic Reviews and Meta-Analyses Extension for Scoping Reviews (PRISMA-ScR) guidelines, a comprehensive search was conducted across Scopus, PubMed, and Web of Science for peer-reviewed, English-language studies published between 2000 and 2024. Eligible studies focused on community knowledge related to malaria prevention, control, and elimination among non-specific populations. Two independent reviewers screened the literature, while a third reviewer resolved any discrepancies. Data were extracted using a standardised framework and analysed systematically in Microsoft Excel.

### Results

The review included 63 studies from 27 countries, predominantly from Africa (68.3%) and Asia (28.6%), with a notable increase in publications in 2021. The findings revealed significant regional variations in community knowledge of malaria transmission, symptoms, and prevention. Higher levels of awareness were consistently linked with better preventive practices and earlier treatment-seeking behaviour. However, critical knowledge gaps persisted, particularly concerning environmental risk

**Data availability statement:** All relevant data are within the paper and its Supporting Information files.

**Funding:** The author(s) received no specific funding for this work.

**Competing interests:** The authors have declared that no competing interests exist.

factors and vector control, highlighting the need for context-specific health education programmes.

## Conclusion

Community knowledge is a pivotal determinant of malaria prevention success, yet disparities persist across endemic regions. To optimise control efforts, it is essential to prioritise locally tailored, evidence-based education that addresses knowledge gaps and strengthens community engagement. Integrating local perspectives into intervention design will be essential for achieving sustainable malaria elimination.

## Introduction

Malaria continues to pose a significant global public health challenge, particularly in sub-Saharan Africa and Southeast Asia, which reported the highest disease burden [1,2]. In 2022, an estimated 249 million cases were reported across 85 endemic countries, marking a 5-million-case increase from the previous year. Since 2015, malaria incidence has risen significantly, with a notable spike between 2019 and 2020, adding 11 million cases to global counts [2]. Although malaria-related deaths decreased slightly from 610,000 in 2021–608,000 in 2022, the persistent transmission underscores the need for intensified and sustained control efforts [2].

Current malaria control efforts primarily focus on biomedical interventions, including prompt diagnosis and treatment, antimalarial drug administration, and vector control measures such as insecticide-treated nets (ITNs), long-lasting insecticidal nets (LLINs), and indoor residual spraying (IRS) [3,4]. While these efforts have yielded substantial progress, malaria remains entrenched in many endemic regions, indicating the need to examine additional factors that influence the effectiveness of control programmes, particularly the critical role of community knowledge and participation [5–7]. A thorough understanding of malaria transmission, symptoms, risk activities, and preventive measures within communities can enhance the effectiveness of interventions, bolster prevention strategies, improve diagnosis and treatment-seeking behaviour, and promote greater community involvement [8,9].

Understanding these non-biomedical influences, it is important to consider the broader social and economic factors that influence health behaviours and access to healthcare services. The lack of resources associated with low socioeconomic status often limits access to malaria prevention and treatment, impacting the timeliness of care-seeking and adherence to preventive behaviours [9,10]. Consequently, individuals in low-income communities often face limited healthcare access, delayed treatment-seeking, and poor adherence to preventive measures, exacerbating malaria persistence [11]. Despite growing recognition of community engagement in malaria control, a significant gap remains in synthesising evidence that describes the extent and nature of community knowledge and how it is reported about malaria programmes. Although some studies have examined community involvement in specific contexts, a comprehensive review of how knowledge, beliefs, and perceptions impact

programme effectiveness across diverse settings is still needed [12–14]. Addressing persistent misconceptions about malaria is essential for creating interventions that are both scientifically rigorous and culturally relevant, thereby enhancing their effectiveness and sustainability [8,15].

To date, no comprehensive review has gathered evidence worldwide on the role of community knowledge in malaria control. This scoping review aims to fill this gap by synthesising and charting the existing literature on the role of community knowledge in malaria prevention, control, and elimination. In particular, it seeks to identify key themes, challenges, and the potential for incorporating community knowledge into interventions. The findings are intended to assist policy-makers, researchers, and practitioners in adapting to their local settings and promoting more effective and sustainable strategies for malaria elimination. Overall, this review will provide a comprehensive overview of the role of community knowledge, highlighting the necessity for culturally appropriate approaches in global malaria efforts, and will guide future research and policy.

## Methodology

### Materials and methods

This study is based exclusively on the analysis of secondary data and therefore does not require approval from a Human Research Ethics Committee. Nevertheless, it has been duly registered with the National Medical Research Register (NMRR ID-23-03703-VZJ). The full methods protocol for this study has been previously published [16]. The Preferred Reporting Items for Systematic Reviews and Meta-Analyses extension for Scoping Reviews (PRISMA-ScR) was used in the design and conduct of this scoping review (S1 Checklist). It utilises a 22-item checklist to ensure high standards in defining the search strategy, eligibility criteria, screening strategy, and data organisation [17]. A comprehensive literature search was performed in the peer-reviewed databases, emphasising community knowledge in malaria prevention, control, and elimination programmes. The review adheres to Joanna Briggs Institute (JBI) protocols [18], which utilise the Arksey and O'Malley framework [19], further adapted by Levac et al. [20]. As a scoping review, it broadly synthesises available evidence, charts methodologies, and identifies research gaps without performing an in-depth critical analysis [21,22].

### Search strategy

The databases Scopus, PubMed, and Web of Science were systematically searched using a specific set of keywords: "community knowledge" OR "public knowledge," "population knowledge," OR "community awareness" OR "public awareness" OR "population awareness" OR "educational community" OR "public education" OR "population education" AND "prevention" OR "control" OR "elimination" AND "malaria." The search was limited to the titles and abstracts of publications from January 2000 to December 2024, ensuring a focused review of the literature published within this period.

Comprehensive database coverage is essential in this review. Scopus, PubMed, and Web of Science were selected for their broad and complementary coverage of biomedical, public health, and multidisciplinary research. PubMed offers comprehensive coverage of clinical and health sciences, while Scopus and Web of Science extend coverage to global health, environmental studies, and social science domains relevant to the review objectives. To balance comprehensiveness with feasibility and minimise duplication, the search strategy was limited to these core databases. Additionally, reference lists of all included studies were manually screened to identify additional eligible articles not captured during the database search. Although not included in the current review, the use of supplementary sources such as Embase or region-specific databases may further improve coverage and will be considered in future reviews.

### Eligibility criteria

Studies were included if they met the inclusion criteria such as original research published in English, using appropriate methodologies (qualitative, quantitative, or mixed-methods designs), case studies, project reports, or programme

evaluations. Additionally, studies were required to provide insight into the role of community knowledge in malaria prevention, control, or elimination. Exclusion criteria included systematic reviews, other reviews, non-English publications, studies unrelated to malaria, book chapters, abstracts, conference proceedings, and articles lacking methodological clarity. Studies involving modelling, simulation, prediction, or machine learning will also be excluded.

### Study selection

Two reviewers conducted a thorough literature search across designated databases, removing duplicate records prior to screening. The screening process unfolded in two stages: title-abstract screening and full-text screening. A pilot sample of 20 articles was used to train the screeners, with a concordance rate exceeding 90% indicating their readiness to proceed. Two reviewers independently evaluated each record for inclusion or exclusion in both stages, while a third reviewer resolved any disagreements. During the title-abstract screening phase, articles were required to meet the primary inclusion criteria and at least one secondary criterion. In the full-text screening phase, articles were required to satisfy all inclusion criteria, with none of the exclusion criteria being applicable. Additionally, the bibliographies of the included articles were reviewed to expand the search. If a full-text article was not accessible, the primary author contacted the corresponding author to request access.

### Data extraction

The data were extracted using a standardised data extraction framework to systematically capture the key attributes of published research literature, facilitating efficiency and consistency throughout the process. This framework collected essential bibliographic details, including article title, authors, publication years, study aims, study sites (countries), study populations, methodologies, sample sizes, languages, and specific malaria types under investigation (prevention, control, elimination). Together, these components provided a comprehensive overview of research data, supporting subsequent data analysis. Reviewers conducted a pilot study to refine the framework's labels, resolve any disagreements, and improve inter-reviewer reliability. Each article underwent a thorough peer review by two reviewers, with a third reviewer adjudicating any disagreements. The data will be imported into Microsoft Excel for analysis and summarisation. This review did not extract or evaluate data on the effectiveness or impact of community knowledge, but focused on describing the scope, content, and themes present in published literature.

### Result

A total of 842 articles were initially identified across three databases (Fig 1). After removing 86 duplicates, 756 articles remained and were screened based on their titles and abstracts, leading to the exclusion of 599 articles. Subsequently, 157 full-text articles were assessed for eligibility, of which 94 were excluded: 89 were unrelated to the scope of this review, and 5 were unavailable for retrieval. Ultimately, 63 articles met the inclusion criteria and were selected for final evaluation (Table 1). Although this review included 63 studies, not all assessed every aspect of community knowledge. Specifically, knowledge of malaria transmission was evaluated in 54 studies, knowledge of malaria symptoms in 40 studies, and knowledge of malaria prevention or control measures in 53 studies. The scope and thematic emphasis varied across studies, reflecting differences in focus and research objectives.

Geographically, the majority of studies originated from Africa (n = 43, 68.3%), followed by Asia (n = 18, 28.6%), with the Americas contributing a smaller proportion (n = 2, 3.2%). This scoping review encompasses studies from 27 countries, providing a comprehensive global perspective on malaria research (Fig 2). Africa had the highest representation, with 18 countries, followed by Asia (n = 8), while the Americas were represented by a single country (Colombia). The most frequently studied countries were Ethiopia (n = 9), Tanzania (n = 7), India (n = 7), and South Africa (n = 6). The retrieved publications span over two decades (2000–2024), with the highest number of studies published in 2021 (n = 7), followed by 2014, 2017, and 2023 (n = 6 each), reflecting periods of increased research activity in malaria-related studies (Fig 3).

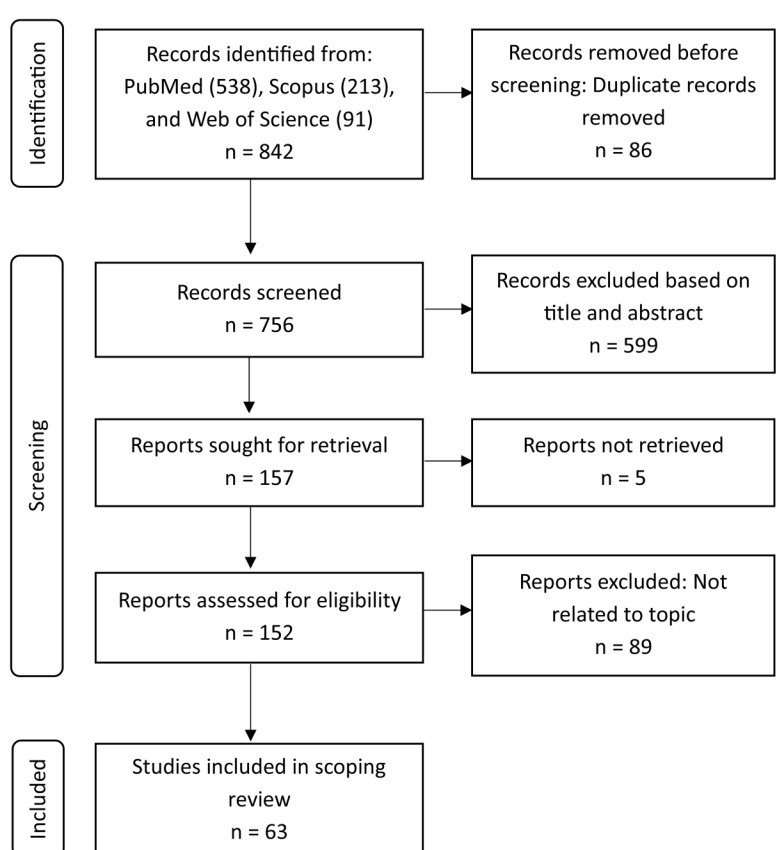

**Fig 1. Flow chart of literature search for the impact of community knowledge in malaria programmes according to the PRISMA-ScR.**

## Community knowledge of malaria transmission

Community knowledge of malaria transmission varied across regions, with 54 studies examining this aspect. The proportion of respondents correctly identifying mosquitoes as the primary vector ranged from 48.8% in Equatorial Guinea [23] to 100% in Ethiopia [24], highlighting disparities in public awareness. Knowledge of mosquito breeding sites also differed significantly. While 87.8% and 79.1% of respondents in Ethiopia [25,26] and 72.5% in Iran [27] correctly identified stagnant water as a breeding site, knowledge was considerably lower in Nepal (59.8%) [28] and Eritrea (15%) [29]. These findings indicate a pressing need for targeted education on environmental risk factors associated with malaria. Knowledge of mosquito biting times was generally high, with 74.7% to 89.9% of respondents in Eritrea [29], Ethiopia [26], and Equatorial Guinea [23] correctly recognising peak mosquito biting hours (Table 2). However, the regional variations in awareness suggest the necessity for ongoing community-based education programmes to ensure an accurate understanding of mosquito behaviour and malaria prevention strategies.

## Community knowledge of malaria symptoms

Community knowledge of malaria symptoms was assessed in 40 studies, with fever being the most commonly recognised symptom, evaluated in 35 of those studies. The prevalence varied from 6.5% in Ethiopia [30] to 95.1% in Tanzania [31].

**Table 1. Summary of articles that met the inclusion criteria and were included in the final analysis.**

| # | Authors (Year) | Country | Study design | Sample size | Community Knowledge of Malaria | The role of community knowledge |
|---|---|---|---|---|---|---|
| 1 | Govere et al. (2000) | South Africa | Cross-sectional | 299 | **Transmission:** -Mosquito bite (65.9%). **Symptom:** -Shivering (63.2%) and fever (31.8%). **Preventive/control:** -Mosquito coil (28.4%) and repellents (16.7%). | -Integrated malaria control activities into the broader priorities of the community. -Ongoing awareness initiatives to solicit and sustain community interest. |
| 2 | Philip B. Adongo (2005) | Ghana | Mixed methods, including FGD | 150 | **Transmission:** -Mosquito bite (79%). **Symptom:** -Hot body (97%), vomiting (38%), chills (27%), and headache (26%). | -Sustain the use of ITNs in the community. -Improve perceptions of ITNs as a malaria prevention tool to enhance their uptake and use. -Tailor health education to distinguish clinical malaria fever from common fever within the local context. |
| 3 | Sanjana et al. (2006) | Indonesia | Random sampling | 1,000 | **Transmission:** -Mosquito bite (69%). **Symptom:** -Fever and chills (93%), nausea and vomiting (17%), and headache (7%). **Preventive/control:** -Keep the house clean (54.7%), bed nets (13.9%), and IRS (12.7%). **Treatment:** -Primaquine (73%). | -Increased community awareness to improve treatment-seeking behaviour. -Emphasized the need for community-specific information on malaria to enable policymakers and programme managers to design an effective malaria control programme. |
| 4 | Ajayi et al. (2008) | Nigeria | Comparative study (intervention vs control) | 611 | **Transmission:** -Mosquito bite (76%). **Preventive/control:** -Chemoprophylaxis (72.5%), bed nets (12.2%), and insecticides (1.2%). **Treatment:** -Antimalarial drugs (73.3%). | -The use of the guideline with adequate training significantly improved the correctness of malaria treatment with chloroquine at home. -Adoption of this mode of intervention is recommended to improve compliance with drug use at home. |
| 5 | Joshi et al. (2008) | Nepal | Cross-sectional | 1,330 | **Transmission:** -Mosquito bite (72.6%), and stagnant water as a mosquito breeding place (59.8%). **Symptom:** -Fever with chills (50.4%). **Preventive/control:** -Bed net (92%), cleaning environment (22.9%), and insecticide spraying (18.9%). | -Most households use bed nets as they know it can prevent mosquito bites. -Improve education and effective messages related to the mosquito breeding places and their role in disease transmission. -Informed the community about the availability of free treatment for malaria. -Health education materials aim to increase knowledge and practice to prevent and control malaria infection. |
| 6 | A. L. Malisa (2009) | Tanzania | Mixed quantitative and qualitative (FGD) | 200 | **Transmission:** -Mosquito bite (75%). **Preventive/control:** -ITNs (65%) and mosquito nets (12.5%). | -Implement effective, appropriate, and sustainable malaria interventions. -Improvement by focusing on factors contributing to low mosquito net coverage. |
| 7 | Hlongwana et al. (2009) | Swaziland | Cross-sectional | 320 | **Transmission:** -Mosquito bite (99.7%). **Symptom:** -Fever (75%), chills (70%), and headache (72%). **Preventive/control:** -Mosquito net (23.1%) and mosquito coils (13.1%). | -Need to improve the availability of information through the preferred community channels. |

*(Continued)*

| # | Authors (Year) | Country | Study design | Sample size | Community Knowledge of Malaria | The role of community knowledge |
|---|---|---|---|---|---|---|
| 8 | Legesse et al. (2009) | Ethiopia | Cross-sectional | 770 | **Transmission:**<br>-Mosquito bite (81.6%) and biting time (42.6%).<br>**Symptom:**<br>-Shivering (54.3%), sweat, and fever (6.5%).<br>**Preventive/control:**<br>-Bed net (76.0%).<br>**Treatment:**<br>-Antimalarial drugs (81.6%). | -Emphasising health education programme for prevention and control.<br>-Improved communities' treatment-seeking behaviour. |
| 9 | Kinung'Hi et al. (2010) | Tanzania | Cross-sectional | 504 | **Transmission:**<br>-Mosquito bite (92.1%).<br>**Symptom:**<br>-Fever (86.5%) and vomiting (60.8%).<br>**Preventive/control:**<br>-Insecticide used to impregnate nets (78.8%). | -Health facilities are the right place for malaria treatment.<br>-Improved health education campaigns.<br>-Increase awareness of ITNs or mosquito net usage. |
| 10 | Sood et al. (2010) | India | Cross-sectional | 596 | **Transmission:**<br>-Mosquito bite (80%).<br>**Symptom:**<br>-Fever with chills and shivering (82%).<br>**Preventive/control:**<br>-Bed net (12%) and LLINs (98.3%). | -LLINs are widely accepted in the user communities and are the safest method. |
| 11 | Das et al. (2011) | India | Cross-sectional | 300 | **Transmission:**<br>-Mosquito bite (72.3%).<br>**Symptom:**<br>-Fever (93.7%), chills (85.5%), headache (55.3%), body ache (43.7%), and vomiting (18%).<br>**Preventive/control:**<br>-Mosquito control and personal protection (70.3%). | -Communication media need to be more women-centric.<br>-Strengthen public awareness.<br>-Regular follow-up monitoring to ensure the information leads to behaviour change. |
| 12 | Kumbulani W. Hlongwana (2011) | South Africa | Cross-sectional | 602 | **Transmission:**<br>-Mosquito bite (84.6%).<br>**Symptom:**<br>-Knowledge of signs and symptoms is very low.<br>**Preventive/control:**<br>-IRS (70%). | -Enhance the Malaria Control Programme by improving the dissemination of malaria-related information, education, and communication strategies. |
| 13 | Mussa Soleimani Ahmadi, (2012) | Iran | Intervention community-based | 592 | -Knowledge regarding malaria transmission increased from 77.5% to 90.3% (Intervention arm) and from 69.4% to 87.9% (Control arm).<br>-Knowledge of mosquito breeding improved, with a slightly higher increase in the control group.<br>-The educational program increased bed net usage to 92.5% in the intervention arm and 87.1% in the control arm. | -Improve the use of LLINs.<br>-Increase motivation to use bed nets. |
| 14 | Abate et al. (2013) | Ethiopia | Cross-sectional | 284 | **Transmission:**<br>-Mosquito bite (85.2%).<br>**Symptom:**<br>-Fever (94.4%), chills (93.3%), back pain/joint pain (69.4%), and nausea/vomiting (66.6%).<br>**Preventive/control:**<br>-Mosquito net (93.7%), drain stagnant water (84.2%), and IRS (78.9%). | -Raise community awareness about the proper uses of preventive measures. |

*(Continued)*

**Table 1.** (Continued)

| # | Authors (Year) | Country | Study design | Sample size | Community Knowledge of Malaria | The role of community knowledge |
|---|---|---|---|---|---|---|
| 15 | Das et al. (2013) | India | Qualitative study | 272 | **Transmission:**<br>-Malaria was transmitted through mosquito bites.<br>-Rains lead to more mosquito breeding sites and hence more malaria cases.<br>**Symptom:**<br>-The majority perceived feeling cold, shivering, fever, intermittent fever, vomiting, and headache.<br>**Preventive/control:**<br>-Fumigating the house in the evenings with dried leaves, husk, straw, or firewood was reported to be the most common way of avoiding mosquitoes.<br>-Most were aware that mosquito nets can prevent malaria. | -Malaria control has advanced with key strategies like Artemisinin-based Combination Therapies (ACTs) and LLINs, enhancing prevention and treatment.<br>-Strengthen the design, implementation, and monitoring of community-based malaria control interventions to enhance their effectiveness and sustainability. |
| 16 | Ediau et al. (2013) | Uganda | Cross-sectional | 770 | **Preventive/control:**<br>-IRS (61.4%). | -Effective community engagement and awareness campaigns for the success of the IRS programme. |
| 17 | Gobena et al. (2013) | Ethiopia | Cross-sectional | 2,867 | **Transmission:**<br>-Mosquito bite (56.1%).<br>**Preventive/control:**<br>-Mosquito nets (82.2%) and IRS (52.2%). | -Correct knowledge of the disease and tailored behavioural change communications are needed.<br>-Further efforts in increasing IRS coverage to maximize the benefit of the intervention. |
| 18 | Anand T. (2014) | India | Cross-sectional | 100 | **Transmission:**<br>-Mosquito breeds in stagnant water (68%).<br>**Preventive/control:**<br>-Liquid vaporizer (60%) and insecticidal spray (32.2%). | -Develop targeted strategies to effectively disseminate information on mosquito vector control and prevention to the population. |
| 19 | Forero et al. (2014) | Colombia | Cross-sectional | 267 | **Transmission:**<br>-Mosquito bite (85.4%).<br>**Symptom:**<br>-Fever (52.4%), headache (16.9%), and chills (16.1%).<br>**Preventive/control:**<br>-ITNs (71.9%) and IRS (33%). | -High usage of ITNs due to the respondents knowing malaria transmission through mosquitoes.<br>-Having a good knowledge of the symptoms encourages seeking early treatment. |
| 20 | Mboera et al. (2014) | Tanzania | Cross-sectional | 962 | **Transmission:**<br>-Mosquito bite (94.8%) and mosquito breeding site (69.9%).<br>**Preventive/control**:<br>-Larviciding will reduce larvae (92.3%) and the risk of getting malaria (91.9%). | -Supports a favourable environment for implementing larviciding as a malaria control measure in rural areas. |
| 21 | Obembe et al. (2014) | Nigeria | Cross-sectional | 280 | **Transmission:**<br>-Mosquito bite (93%) and mosquito breeding sites in stagnant water (69%).<br>**Symptoms:**<br>-High fever (46%) and cold/chills (20%).<br>**Preventive/control:**<br>-Treated net (60%), insecticide spray (54%), and mosquito coils (48%).<br>**Treatment:**<br>-Artesunate (34%), chloroquine (10%), and ACTs (6.1%). | -Strengthen community outreach by improving the quality and clarity of malaria prevention messages.<br>-The use of at least one mosquito control method is likely due to increased awareness of malaria prevention and the role of mosquitoes in its transmission. |

*(Continued)*

| # | Authors (Year) | Country | Study design | Sample size | Community Knowledge of Malaria | The role of community knowledge |
|---|---|---|---|---|---|---|
| 22 | Soleimani-Ahmadi et al. (2014) | Iran | Cross-sectional | 400 | **Transmission:** -Mosquito bite (77.8%) and mosquito breeding sites in stagnant water (72.5%). **Symptoms:** -Fever and chill (68.7%) and fever (16.7%). **Preventive/control:** -LLINs (60.8%), IRS (15%), and mosquito coils (48%). | -Increase awareness of vector control to reduce mosquito population density. -Effective educational programmes contributed to the increased use of LLINs. |
| 23 | Syed Masud Ahmed et al. (2014) | Bangladesh | Cross-sectional | 750 | **Transmission:** -Mosquito bite (95.7%). **Symptom:** -Know the common malaria symptoms (94.3%). **Preventive/control:** -Insecticide nets (74.4%). **Treatment:** -Malaria is treated by allopathic medicine (99.3%). | -Enhancing malaria awareness through interpersonal communication. |
| 24 | Gutasa et al. (2015) | Ethiopia | Cross-sectional | 283 | **Transmission:** -Mosquito bite (87%). **Symptom:** -Fever (93.7%), headache (84.3%), and coldness/shivery (77.6%) **Preventive/control:** -Bed nets (85.5%). | -Appropriate health education among communities about malaria information. -Increase community mobilization to create awareness of the appropriate utilization of bed nets. |
| 25 | Tarimo D.S. (2015) | Tanzania | Cross-sectional | 1,880 | **Symptom:** -Fever (95.1%) and shivering (57.9%). **Preventive/control:** -ITNs (93.3%) and LLINs (90%). **Treatment:** -Artemether-lumefantrine (AL) (79.7%). | -Communities clearly understand that mosquitoes transmit malaria. -Communities were aware that ITNs prevent mosquito bites and reduce malaria risk. |
| 26 | Muriuki et al. (2016) | Kenya | Cross-sectional | 334 | **Transmission:** -Mosquito bite (97%). **Symptom:** -Fever (93.7%), headache (67.4%), chills/shivering (64.7%). **Preventive/control:** -Mosquito nets (79.6%), environmental clearing (20.7%), and IRS (3%). **Treatment:** -Artemether-lumefantrine (AL) (24.3%). | -Mosquito nets were widely used with higher ownership and usage to protect from mosquito bites. -Increase awareness of proper diagnosis and management of malaria disease. |
| 27 | Romay-Barja et al. (2016) | Equatorial Guinea | Cross-sectional | 428 | **Transmission:** -Mosquito bite (48.83%) and biting time (89.95%). **Symptom:** -Fever (75.93%). **Preventive/control:** -Bed net (45.79%). **Treatment:** -Artemether (25.7%). | -Empowering adequate knowledge aimed at behavioural changes to improve disease prevention. |

*(Continued)*

**Table 1.** (Continued)

| # | Authors (Year) | Country | Study design | Sample size | Community Knowledge of Malaria | The role of community knowledge |
|---|---|---|---|---|---|---|
| 28 | Seyoum et al. (2016) | Ethiopia | Cross-sectional | 2,319 | **Transmission:**<br>-Mosquito bite (95%).<br>**Symptom:**<br>-Fever (79.6%), chills (48.6%), and severe headache (38.1%).<br>**Preventive/control:**<br>-ITNs (40.4%). | -Strengthen overall knowledge about malaria, especially on the mode of disease transmission, proper handling, and effective use of ITNs. |
| 29 | Shanfeng Tang et al. (2016) | China | Cross-sectional | 1,321 | -Individuals with a history of malaria or exposure to information via community or electronic media were more likely to have greater knowledge. | -Determinants factors for public knowledge of Malaria during the National Malaria Elimination Programme.<br>-Improve malaria promotion strategies for malaria prevention. |
| 30 | Chantal Marie Ingabire et al. 2017) | Rwanda | Interventional | 320 participants for quantitative and 45 participants for qualitative | **Transmission:**<br>-Mosquito bite (91.9%) and mosquito breeding sites (72.4%).<br>**Symptom:**<br>-Aware of three or more malaria symptoms (70.3%).<br>**Preventive/control:**<br>-Mentioned at least three effective methods to prevent malaria (61.3%). | -Actively engage in Bti activities for malaria control, with 72.5% of participants expressing willingness to dedicate at least one hour per day to support future Bti applications.<br>-Knowledge of four malaria symptoms increased odds 3.115. |
| 31 | Gloria Isabel Jaramillo-Ramirez (2017) | Colombia | Cross-sectional | 120 | **Transmission:**<br>-Mosquito bite (65%).<br>**Symptom:**<br>-Fever (72.5%), headache (55%), and chills (50%).<br>**Preventive/control:**<br>-Spraying insecticide (90%) and bed nets (59%). | -Strengthen health education programs to raise awareness and promote effective malaria prevention measures.<br>-Enhance access to knowledge and promote health-seeking behaviours for early malaria treatment. |
| 32 | Manana et al. (2017) | South Africa | Cross-sectional | 400 | **Transmission:**<br>-Mosquito bite (99%).<br>**Symptom:**<br>-Headache (78%), fever (54%), feeling cold (60%), and vomiting (30%).<br>**Preventive/control:**<br>-IRS (75%), mosquito coils (55%), mosquito repellent (14%), and insecticide (7%).<br>-Sterile Insect Technique (SIT) as a vector control tool. | -The community sought treatment within 24 hours after having experienced any symptoms of malaria.<br>-The community supported passive vector control strategies such as IRS and SIT as supplementary passive vector control interventions. |
| 33 | Moshi et al. (2017) | Tanzania | Purposive study | 128 | **Transmission:**<br>-Transmission might occur indoors and outdoors.<br>-Mosquito biting rates were higher during the rainy season, attributed to the increase in available breeding sites.<br>**Risk activities:**<br>-Cooking, eating outdoors (dinner), washing kitchen utensils, and conversing with other family members.<br>**Preventive/control:**<br>-The majority (>95%) of participants indicated that bed nets were their communities' most common and widely used indoor mosquito control intervention. | -Promoting outdoor malaria control measures requires advocacy on evolving transmission patterns alongside scaling-up efforts.<br>-Provide communities with accurate, up-to-date malaria information.<br>-Educational campaigns on outdoor malaria transmission and interventions are vital for effective control. |

*(Continued)*

| # | Authors (Year) | Country | Study design | Sample size | Community Knowledge of Malaria | The role of community knowledge |
|---|---|---|---|---|---|---|
| 34 | Naing et al. (2017) | Myanmar | Cross-sectional | 6,597 | **Transmission:**<br>-Mosquito bite (87%).<br>**Preventive/control:**<br>-Mosquito nets (76%) and ITNs (28.9%). | -Those having poor knowledge about malaria have poor treatment-seeking behaviour.<br>-A need to promote awareness about the role of early diagnosis and appropriate treatment. |
| 35 | Sarah N. Cox (2017) | South Africa | Intervention (Quasi-experimental) | 1,330 | -Higher knowledge among the intervention arm vs the non-intervention arm.<br>-Improvement in malaria knowledge pre- and post-by 9% in the intervention arm. | -Improvement of knowledge to increase uptake of malaria control interventions.<br>-Post-intervention, participants demonstrated improved knowledge and expressed that the program was valuable and rewarding. |
| 36 | Alelign et al. (2018) | Ethiopia | Two-stage random cluster | 144 | **Transmission:**<br>-Mosquito bite (78.5%).<br>-Plasmodium as the causative agent (24.3%).<br>**Symptom:**<br>-Fever (34%) and shivering (25%).<br>**Preventive/control:**<br>-Draining logged water (57.6%), environmental clearing (27.1%), and ITNs (100%). | -Community involvement in malaria control activities.<br>-Strengthened the awareness campaign. |
| 37 | Finda et al. (2019) | Tanzania | Mixed methods | 307 | **Transmission:**<br>-Mosquito as the vector and Role of mosquito swarms in malaria transmission (67.9%). | -Swarm-targeting interventions for the control of malaria vectors and transmission. |
| 38 | Magaço et al. (2019) | Mozambique | Cross-sectional | 190 | **Preventive/control:**<br>-IRS (48.94%). | -Improved IRS acceptance in poor and rural areas.<br>-Community leader involvement in IRS promotion. |
| 39 | Nganga et al. (2019) | Kenya | Cross-sectional | 80 | **Preventive/control:**<br>-Mosquito net (64.3%), treated mosquito net (30.4%), preventive medicine (32.1%), and filling breeding sites (68.0%). | -Communities recognized treated mosquito nets as an effective measure for preventing malaria transmission.<br>-Emphasize awareness creation, education, and empowering communities to use available vector control measures, including house screening for better protection. |
| 40 | Omonijo et al. (2019) | Nigeria | Cross-sectional | 352 | **Transmission:**<br>-Proper knowledge of malaria transmission (96.3%). | -Enhance education and awareness in the communities on the proper and correct usage of the nets. |
| 41 | Aung PL et al. (2020) | Myanmar | Cross-sectional | 250 | −38.4% of respondents had good knowledge, 56.8% had a good attitude, and 21.6% had a good practice of malaria. | -Community involved in health education programmes.<br>-Malaria intervention approaches should be simple, user-friendly, and presented in concise, easy-to-understand formats. |
| 42 | Singh et al. (2020) | India | Cross-sectional | 1,470 | **Transmission:**<br>-Mosquito bite (88%).<br>**Symptom:**<br>-Fever (84.1%), chills & rigor (75.2%), headache (54.2%), and vomiting (30.5%).<br>**Preventive/control:**<br>-Cover the body with clothes (32.1%), bed nets (36.2%), and repellent (9.7%). | -Good knowledge influenced practicing modern malaria preventive measures.<br>-Students can act as powerful agents of awareness programmes in the community. |

*(Continued)*

**Table 1.** (Continued)

| # | Authors (Year) | Country | Study design | Sample size | Community Knowledge of Malaria | The role of community knowledge |
|---|---|---|---|---|---|---|
| 43 | Tsigie Baye Aragie (2020) | Ethiopia | Cross-sectional with mixed quantitative and qualitative study | 766 | **Transmission:**<br>-Mosquito breeds in stagnant water (87.8%).<br>**Preventive/control:**<br>-LLINs (84.7%) and IRS (83.7%). | -Benefits and improved use of LLINs and IRS as malaria prevention activities.<br>-Focus on targeted community (socio-demographic factors) to improve awareness of the malaria prevention programme. |
| 44 | Asale et al. (2021) | Ethiopia | Qualitative study | 3,010 | **Transmission:**<br>-Mosquito bite (100%).<br>**Prevention/control:**<br>-Use LLINs and IRS in some health clusters.<br>-Remove mosquito breeding habitat by filling and draining stagnant water. | -Advocate for the community on knowledge and correct usage of LLINs. |
| 45 | Menjetta T. (2021) | Ethiopia | Cross-sectional | 421 | **Transmission:**<br>-Mosquito bite (63.4%), mosquito breeding site in stagnant water (79.12%), and mosquito biting time at night (82.7%).<br>**Symptom:**<br>-Fever (49.6%), headache (40.8%), and chills (26.8%). | -Knowledge about mosquito bites at night encourages the community to take appropriate preventive measures and properly use mosquito nets.<br>-The necessity of health education to raise the community's awareness about malaria. |
| 46 | Munajat et al. (2021) | Malaysia | Cross-sectional | 536 | **Transmission:**<br>-Mosquito bite (98.6%).<br>**Symptom:**<br>-Fever (40.6%), headache (14.4%), and rigor/chills (14.1%).<br>**Preventive/control:**<br>-Bed nets (50.2%) and insecticide spraying (19.1%). | -Improve community awareness about malaria vectors, mosquito behaviour, and breeding sites.<br>-Improving ownership and utilization of bed nets.<br>-Treatment-seeking behaviour to seek clinical treatment for malaria.<br>-Enhancing malaria surveillance, vector control (IRS and LLINs), education, and information campaigns. |
| 47 | Munzhedzi et al. (2021) | South Africa | Mixed method | 261 | **Transmission:**<br>-Mosquito bite (95%).<br>**Symptom:**<br>−99.2% correctly identified at least one of the 2 common symptoms.<br>−21% were able to identify all 3 common symptoms correctly.<br>**Preventive/control:**<br>-Wearing long-sleeved clothes (39.1%) and bed nets (23.8%). | -Need to distribute bed nets to the community. |
| 48 | Omotayo et al. (2021) | Nigeria | Cross-sectional | 502 | **Preventive/control:**<br>-Insecticides (77.7%) and LLINs (48.2%). | -Need to orient residents on the positive impacts of LLINs and downplay the inconvenience suffered by using LLINs. |
| 49 | Rajvanshi et al. (2021) | India | Cross-sectional | 773 | **Transmission:**<br>-Mosquito bite (78%).<br>**Symptom:**<br>-Fever with chills (65.1%), headache (29.5%), and high fever (29.2%).<br>**Preventive/control:**<br>-LLINs (41.1%) and IRS (38.8%).<br>**Treatment:**<br>-Chloroquine (5%). | -Improvement in respondents' self-reported knowledge of malaria.<br>-Improved health literacy among communities.<br>-Significant reduction of indigenous malaria cases. |

*(Continued)*

| # | Authors (Year) | Country | Study design | Sample size | Community Knowledge of Malaria | The role of community knowledge |
|---|---|---|---|---|---|---|
| 50 | Wang et al. (2021) | Sierra Leone | Cross-sectional | 300 | **Transmission:**<br>-Mosquito bite (86.6%).<br>**Symptom:**<br>-Fever (44.3%), body ache (38.9%), and loss of appetite (36.6%).<br>**Preventive/control:**<br>-Treated bed nets (75.6%), keep the surrounding clean (32.1%), and IRS (2.7%). | -Existing knowledge of malaria should be sustained and reinforced, and the availability and use of malaria prevention measures should be promoted.<br>-Enhance the awareness and correct use of protective measures. |
| 51 | Azoukalne Moukenet (2022) | Chad | Cross-sectional study | 278 | **Transmission:**<br>-Mosquito bite (68.3%).<br>**Symptom:**<br>-Fever (81.7%), vomiting (75.2%), and chills (25.3%).<br>**Preventive/control:**<br>-LLINs (79.9%). | -Increased knowledge and access to LLINs used to reduce malaria morbidity.<br>-Prioritize providing access to LLINs for specific ethnic groups with limited knowledge and low utilization of LLINs. |
| 52 | Djoufounna et al. (2022) | Cameroon | Cross-sectional | 413 | **Transmission:**<br>-Mosquito bite (94.7%).<br>**Preventive/control:**<br>-Mosquito net (92.3%) and insecticide spray (20.3%). | -Good attitudes and practices of LLINs usage as the main preventive measure.<br>-Strengthened awareness of malaria prevention and treatment. |
| 53 | Hasabo et al. (2022) | Sudan | Cross-sectional | 310 | **Transmission:**<br>-Mosquito bite (86.8%) and *Plasmodium* as the causative agent (20%).<br>**Symptom:**<br>-Fever with shivering (75.8%).<br>**Preventive/control:**<br>-Mosquito coils/repellent (35%), ITNs (60.2%), and IRS (42.4%). | -Seek early treatment when they get symptoms.<br>-Enhance and complement malaria knowledge with effective preventive and control strategies. |
| 54 | Kaboré et al. (2022) | Burkina Faso | Cross-sectional | 1,394 | **Knowledge (FGD)**<br>-Most of the respondents had good knowledge.<br>-Fever is the main symptom of malaria.<br>-Mosquitoes transmit malaria.<br>-Preventive measures against malaria are well identified (bed nets, water management). | -Combined with the good knowledge of malaria disease, it could potentially facilitate a successful implementation of the MFT pilot programme, which is based on activities at the health centre level.<br>-MFT-significant benefits in fighting drug resistance by slowing the fixation of resistant strains and retarding selection pressure to the partner drugs used in artemisinin combinations.<br>-*(MFT- multiple first-line therapies). |
| 55 | Tshivhase et al. (2022) | South Africa | Cross-sectional | 151 | **Transmission:**<br>-Mosquito bite (86%).<br>**Symptom:**<br>-High fever, shivering, and headache (63%).<br>**Preventive/control:**<br>-Mosquito coil and net (38%) and wear long-sleeved clothes (34%). | -Seek treatment within 24 hours after suspecting malaria infection.<br>-Improve the implementation of the prevention and control programme.<br>-Strengthened malaria educational programme for local health authorities. |

*(Continued)*

**Table 1.** (Continued)

| # | Authors (Year) | Country | Study design | Sample size | Community Knowledge of Malaria | The role of community knowledge |
|---|---|---|---|---|---|---|
| 56 | Abdul Rahim et al. (2023) | Malaysia | Cross-sectional | 3085 | **Transmission:**<br>-Mosquito bite (93%).<br>-Correct transmission (53.9%).<br>**Symptom:**<br>-Fever, chills, and rigors (94%).<br>-Correct symptoms (24.1%).<br>**Risk activities:**<br>-Fishing in the swamp (83%) and Recreational activities in the forest (81%).<br>-Correct risk activities (34%).<br>**Preventive/control:**<br>-Antimalarial medication (90%), protective clothing (90%), ITNs (83%), and IRS (76%).<br>-Correct preventive (59.7%). | -Good knowledge of malaria symptoms will encourage communities to seek immediate treatment.<br>-Increased exposure to health education campaigns, the dissemination of information, and enhanced access to healthcare facilities.<br>-Focused research and innovative educational approaches should prioritise addressing the needs of specific vulnerable communities or groups. |
| 57 | Andegiorgish et al. (2023) | Eritrea | Cross-sectional | 380 | **Transmission:**<br>-Mosquito bite (94.2%), biting time (74.7%), and mosquito breeding sites (15%).<br>**Symptom:**<br>-Fever (89.2%) and chills (55.9%).<br>**Preventive/control:**<br>-ITNs (84.4%).<br>**Treatment:**<br>-Anti-malarial prophylaxis (86.3%). | -Communities had a positive attitude towards treatment-seeking behaviour from health facilities.<br>-Communities demonstrated effective practices in malaria prevention. |
| 58 | Dey et al. (2023) | India | Cross-sectional | 456 | **Transmission:**<br>-Mosquito bite (89.7%).<br>**Symptom:**<br>-Fever (89.1%).<br>**Preventive/control:**<br>-LLINS (82.7%) and insecticides (13.8%). | -Good knowledge of malaria within communities has contributed to the widespread usage of LLINs. |
| 59 | Gowelo et al. (2023) | Malawi | Cluster randomized control | 502 | **Transmission:**<br>-Mosquito bite (98%) and malaria caused by female anopheline (58%). | -Knowledge did not help in community participation in Larvae Source Management. |
| 60 | Liheluka et al. (2023) | Tanzania | Qualitative study | 276 | **Transmission:**<br>-Most of the community members have good knowledge of malaria transmission.<br>**Prevention/control:**<br>-Using bed nets will prevent malaria. | -Affordability was a key factor in net ownership and use. Need to distribute the nets freely or subsidised. |
| 61 | Magaco et al. (2023) | Mozambique | Qualitative study | 92 | **Transmission:**<br>-Mosquito bite.<br>**Symptom:**<br>-Chills, joint pain, body aches and headache, fever, dizziness, and vomiting.<br>**Preventive/control:**<br>-Bed nets and cleanliness of the house. | -Strengthen the communication and logistics component of the campaign on net distribution within the community.<br>-Improve community knowledge of malaria using appropriate educational materials and networks.<br>-Improving ownership and the use of insecticide-treated nets. |
| 62 | Naura et al. (2024) | Indonesia | Cross-sectional | 115 | -Knowledge of causes, symptoms, and malaria prevention measures (60%). | -Good knowledge significantly influenced malaria prevention and control efforts. |
| 63 | Tanue et al. (2024) | Cameroon | Cross-sectional | 2,386 | **Transmission:**<br>-Mosquito bite (95.2%).<br>**Symptom:**<br>-Fever (94.4%) and headache (67.7%).<br>**Preventive/control:**<br>-Mosquito nets (92.1%). | -Knowledge of symptoms plays a particular role in seeking healthcare, making early diagnoses, and treating.<br>-Most households practiced bed nets as a malaria prevention method. |

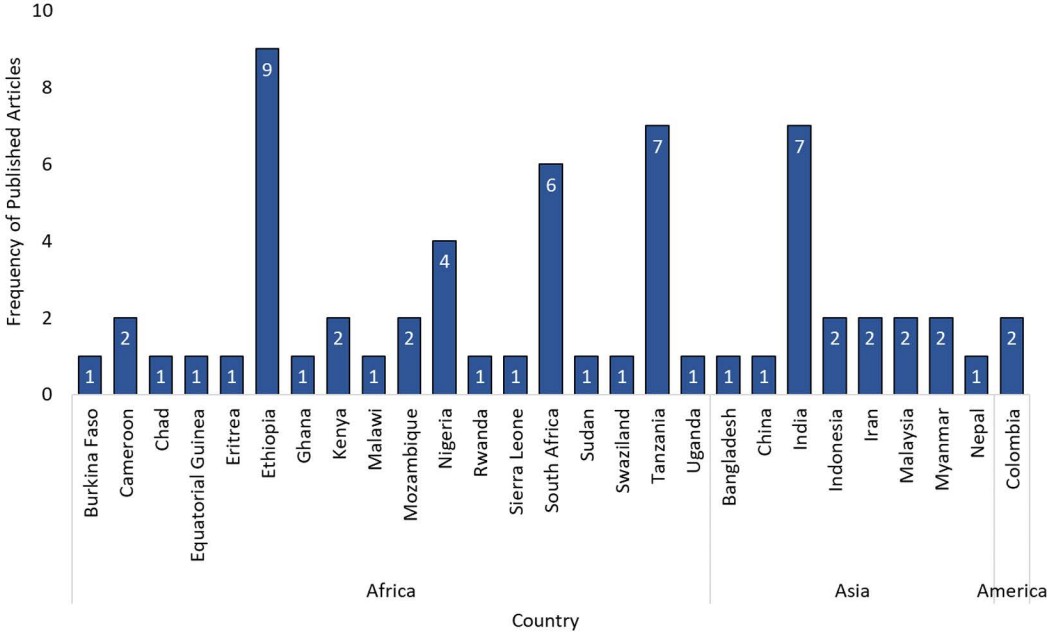

**Fig 2. Distribution of published articles on the impact of community knowledge in malaria programmes by country.**

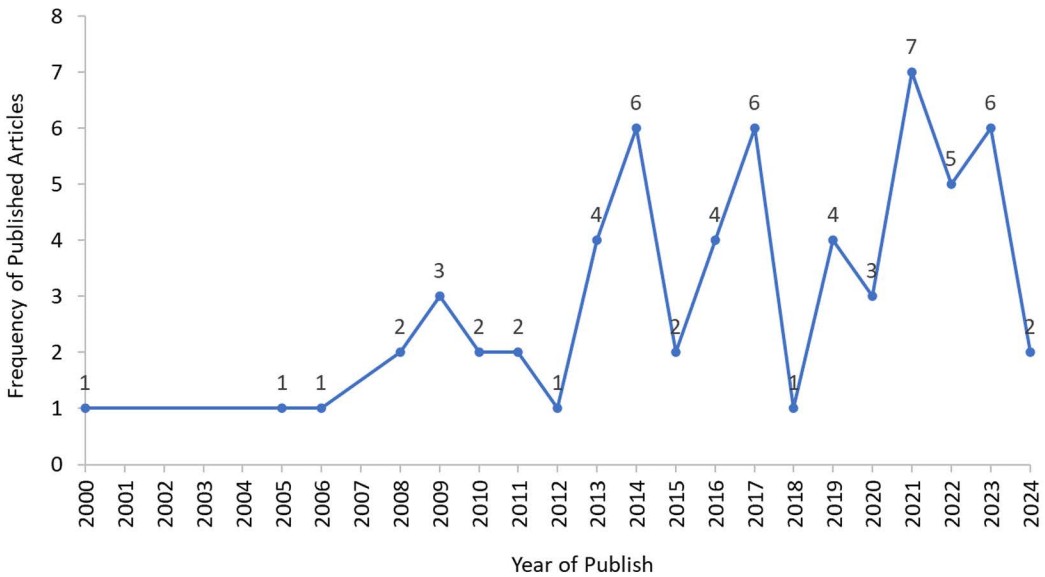

**Fig 3. Distribution of published articles on the impact of community knowledge in malaria programmes by year.**

Other frequently identified symptoms included chills [32,33], headaches [34,35], shivering [36,37], and vomiting [38,39] (Table 2). However, recognition rates differed significantly across countries, underscoring the need for improved health education and awareness campaigns to promote early detection and timely treatment-seeking behaviour.

## Community knowledge of malaria prevention and control

Community knowledge of malaria prevention and control measures was assessed in 53 studies, with a particular emphasis on mosquito nets, ITNs, LLINs, IRS, and environmental management practices. Ethiopia reported the highest knowledge of mosquito nets at 93.7% [40], while India reported the lowest at 12.0% [41]. Similarly, knowledge of ITNs was notably high in Ethiopia (100%) [37] and Tanzania (93.3%) [31], reflecting the higher implementation of vector control measures in these regions. Meanwhile, knowledge of LLINs was high in India (98.3%) [41] and Tanzania (90.0%) [31], while lower knowledge was observed in Nigeria (48.2%) [42] and India (41.1%) [43]. Knowledge of IRS as a malaria prevention strategy showed substantial variation across studies, indicating regional differences in malaria control programme implementation. While knowledge of IRS was notably low in Sierra Leone (2.7%) [44] and Kenya (3.0%) [45], it was significantly higher in Ethiopia (78.9% and 83.7%) [25,40]. Environmental management was also recognised as a crucial malaria prevention strategy, involving various interventions to reduce mosquito breeding sites. These measures include clearing vegetation [37,45], maintaining clean surroundings [44], filling potential breeding sites [46], and draining stagnant water [37,40], all of which help minimise the risk of malaria transmission. Other preventive measures, such as wearing protective clothing [8], using mosquito coils, and repellents [36], were also identified as effective strategies for reducing malaria risk (Table 2). These findings emphasise the importance of integrated vector control strategies and community education in strengthening malaria prevention efforts.

## Community knowledge of malaria treatment

Community knowledge of malaria treatment varied significantly across countries. Artemether-lumefantrine (AL) was the most recognised therapy, reported by 24.3% of respondents in Kenya [45] and 79.7% in Tanzania [31]. Although chloroquine was mentioned less frequently due to resistance, it was still recognised by 5.0% of respondents in India [43] and 10.0% in Nigeria [32], with structured education improving adherence. General awareness of antimalarial drugs was noted by 73.3% of respondents in Nigeria [47], 81.6% in Ethiopia [30], and 90.0% in Malaysia [8], reflecting varying levels of accessibility and health education initiatives. Primaquine awareness was notably high in Indonesia (73.0%), contributing to improved treatment-seeking behaviour and compliance with prescribed treatment regimens [48]. Furthermore, antimalarial prophylaxis was widely recognised in Eritrea (86.3%), correlating with a more proactive approach to healthcare and malaria prevention [29] (Table 2).

## Impact of community knowledge

Findings from various studies highlight the positive impact of increased community awareness on malaria prevention and treatment, while also underscoring the challenges posed by knowledge gaps across different regions. Greater awareness has led to higher adoption of preventive measures, particularly mosquito nets and IRS. This heightened awareness has also encouraged more ownership and utilisation of bed nets, enhancing protection against mosquito bites. Moreover, community knowledge was essential for the success of bed net distribution programmes, with affordability and accessibility identified as crucial factors influencing their uptake [7,45,49,50]. In Mozambique, the active involvement of community leaders in promoting the IRS significantly increased acceptance of the intervention, particularly in rural and lower-income areas [51]. Effective community engagement has been correlated with increased IRS utilisation for malaria prevention. To further optimise the impact of IRS interventions, expanding coverage and accessibility remains essential, as demonstrated by studies in Uganda [52] and Ethiopia [25,53], which emphasise the need for ongoing efforts to strengthen IRS adoption and implementation.

                                                                                   

**Table 2. Summary of community knowledge on malaria transmission, symptoms, prevention/control, and treatment.**

| Country | Community knowledge on malaria | | | |
| --- | --- | --- | --- | --- |
| | **Transmission** | **Symptom** | **Prevention/control** | **Treatment** |
| Equatorial Guinea | -Mosquito bite (48.8%)<br>-Mosquito biting time (89.9%) | -Fever (75.9%) | -Bed nets (45.8%) | -Artemether (25.7%) |
| Ethiopia | -Mosquito bite (56.1%−100%)<br>-Mosquito biting time (42.6%−82.7%)<br>-Mosquito breeding sites (79.1%−87.8%) | -Fever (34.0%−94.4%)<br>-Chills (26.8%−93.3%)<br>-Back pain/joint pain (69.4%)<br>-Nausea/vomiting (66.6%)<br>-Shivering (25.0%−77.6%)<br>-Sweat and fever (6.5%)<br>-Headache (38.1%−84.3%) | -Bed nets (76.0%−85.5%)<br>-Mosquito net (82.2%−93.7%)<br>-Drain stagnant water (57.6%−84.2%)<br>-IRS (52.2%−83.7%)<br>-ITNs (40.4%−100%)<br>-Environmental clearing (27.1%)<br>-LLINs (84.7%) | -Antimalarial drugs (81.6%) |
| Iran | -Mosquito bite (77.8%−90.3%)<br>-Mosquito breeding sites (72.5%) | -Fever (16.7%)<br>-Fever and chills (68.7%) | -Bed nets (87.1%−92.5%)<br>-LLINs (60.8%)<br>-IRS (15.0%)<br>-Mosquito coils (48.0%) | – |
| Nepal | -Mosquito bite (72.6%)<br>-Mosquito breeding sites (59.8%) | -Fever and chills (50.4%) | -Bed nets (92.0%)<br>-Cleaning environment (22.9%)<br>-Insecticide spraying (18.9%) | – |
| Eritrea | -Mosquito bite (94.2%)<br>-Mosquito biting time (74.7%)<br>-Mosquito breeding sites (15.0%) | -Fever (89.2%)<br>-Chills (55.9%) | -ITNs (84.4%) | -Anti-malarial pro-phylaxis (86.3%) |
| Tanzania | -Mosquito bite (75.0%−94.8%)<br>-Mosquito breeding sites (69.9%) | -Fever (86.5%−95.1%)<br>-Vomiting (60.8%)<br>-Shivering (57.9%) | -ITNs (65%−93.3%)<br>-LLINs (90%)<br>-Mosquito nets (12.5%)<br>-Larviciding (91.9%−92.3%) | -Artemether-lumefantrine (AL) (79.7%) |
| India | -Mosquito bite (72.3%−89.7%)<br>-Mosquito breeding sites (68.0%) | -Fever (29.2%−93.7%)<br>-Chills (82%−85.5%)<br>-Chills and rigor (75.2%)<br>-Shivering (82.0%)<br>-Headache (29.5%−55.3%)<br>-Body ache (43.7%)<br>-Vomiting (18.0%−30.5%) | -Bed nets (12.0%−36.2%)<br>-LLINs (41.1%−98.3%)<br>-Insecticide spraying (13.8%−32.2%)<br>-Liquid vaporizer (60.0%)<br>-Repellent (9.7%)<br>-Protective clothing (32.1%)<br>-IRS (38.8%) | -Chloroquine (5%) |
| Nigeria | -Mosquito bite (76.0%−93.0%)<br>-Mosquito breeding sites (69.0%) | -High fever (46.0%)<br>-Chills (20.0%) | -Chemoprophylaxis (72.5%)<br>-Bed nets (12.2%)<br>-ITNs (60.0%)<br>-Insecticide spraying (1.2%−77.7%)<br>-Mosquito coil (48.0%)<br>-LLINs (48.2%) | -Antimalarial drugs (73.3%)<br>-Artesunate (34%)<br>-Chloroquine (10%)<br>-ACTs (6.12%) |
| Sierra Leone | -Mosquito bite (86.6%) | -Fever (44.3%)<br>-Body ache (38.9%)<br>-Loss of appetite (36.6%) | -ITNs (75.6%)<br>-Cleaning environment (32.1%)<br>-IRS (2.7%) | – |
| Kenya | -Mosquito bite (97.0%) | -Fever (93.7%)<br>-Headache (67.4%)<br>-Chills/shivering (64.7%) | -Mosquito net (64.3%−79.6%)<br>-Environmental clearing (20.7%)<br>-IRS (3.0%)<br>-ITNs (30.4%)<br>-Preventive medicine (32.1%)<br>-Filling breeding sites (68.0%) | -Artemether-lumefantrine (AL) (24.3%) |
| Malaysia | -Mosquito bite (93.0%−98.6%) | -Fever (40.6%−94.0%)<br>-Headache (14.4%)<br>-Rigor/chills (14.1%−94.0%) | -Bed nets (50.2%)<br>-Insecticide spraying (19.1%)<br>-Protective clothing (90.0%)<br>-ITNs (83.0%)<br>-IRS (76.0%) | -Antimalarial drugs (90.0%) |
| Indonesia | -Mosquito bite (69.0%) | -Fever and chills (93.0%)<br>-Nausea/vomiting (17.0%)<br>-Headache (7.0%) | -Keep the house clean (54.7%)<br>-Bed nets (13.9%)<br>-IRS (12.7%) | -Primaquine (73.0%) |

Enhanced community knowledge has significantly improved compliance with malaria prevention strategies, particularly LLINs. Evidence from India indicates that increased community knowledge has led to widespread acceptance and proper utilisation of LLINs, contributing to a measurable reduction in malaria morbidity [41,54]. Additionally, education regarding the benefits and correct usage of LLINs has been shown to enhance net utilisation and effectiveness in malaria prevention [24,42,49,55]. In Tanzania [56] and Rwanda [57], active community engagement facilitated the adoption of larviciding as a vector control strategy, reinforcing the importance of integrated malaria prevention measures that combine chemical, biological, and environmental interventions.

Beyond prevention, early malaria treatment is essential for reducing transmission [48,58,59]. Several studies have demonstrated that greater awareness of malaria symptoms leads to earlier healthcare-seeking behaviour, reducing disease progression and transmission risk [8,60–62]. A study in Eritrea found that communities with greater awareness level were more proactive in seeking treatment at health facilities, resulting in higher adherence to medication and improved recovery rates [29]. Conversely, findings from Myanmar indicate that low malaria awareness was directly associated with reduced treatment-seeking behaviour, highlighting the urgent need for ongoing health education initiatives to bridge knowledge gaps and promote early detection and timely treatment [63].

Furthermore, evidence from India suggests that improving health literacy within communities enhances self-reporting of malaria symptoms, contributing to a decrease in indigenous malaria cases [43]. This underscores the urgent need for targeted health education initiatives to bolster awareness and understanding of malaria prevention and control [9,26,64]. Expanding community-wide education is crucial for increasing knowledge about mosquito breeding sites, effective prevention strategies, and malaria control measures [28,30,46]. Emphasis should be placed on the proper use of ITNs and LLINs to ensure their widespread adoption and support malaria prevention efforts [27,65–67]. Moreover, targeted research and educational strategies should prioritise the needs of vulnerable communities where the malaria burden remains high [8].

Effective communication strategies are vital for enhancing malaria awareness and prevention efforts. A study in India indicates that communication media should adopt a more women-centred approach to increase engagement and outreach [39]. Strengthening interpersonal communication has also been identified as a key factor in raising malaria awareness and encouraging proactive health behaviours [68]. To maximise the effectiveness of malaria control programmes, it is essential to improve the dissemination of information through education and communication strategies, ensuring broader accessibility and effective knowledge transfer within communities [69].

## Discussion

This scoping review presents evidence on the role of community knowledge in malaria prevention and control programmes from 2000 to 2024. The majority of studies originated from Africa and Asia, particularly Ethiopia, Tanzania, and India, reflecting the high malaria burden and sustained investments in control strategies within these regions. Notable peaks in publication were observed in 2014, 2017, 2021, and 2023, aligning with intensified global malaria elimination efforts and increased funding [2]. However, the underrepresentation of studies from South America, the Pacific, and parts of Southeast Asia highlights critical gaps in the global evidence base on community-level malaria knowledge.

Although the review primarily focuses on African and Asian contexts, the findings have wider applicability. Common challenges, such as misconceptions, inconsistent use of preventive measures, and delayed treatment-seeking, are also evident in other malaria-endemic regions. For example, similar barriers have been reported among remote and indigenous communities in the Amazon basin [70]. In countries progressing towards elimination, including Malaysia and Iran, community engagement remains central to sustaining gains and preventing resurgence [8,27]. These cross-regional parallels reinforce the global significance of the findings and underscore the importance of context-specific approaches.

Significant disparities persist in community knowledge of malaria transmission and symptoms. For example, while 87.0% of respondents in Ethiopia correctly identified stagnant water as a mosquito breeding site, only 15.0% did so in Eritrea [25,29], indicating uneven dissemination of public health messaging. Fever was the most commonly recognised

symptom across 35 studies, but knowledge of secondary symptoms such as chills, headache, and vomiting was inconsistent. These knowledge gaps may delay care-seeking and exacerbate disease severity. Comprehensive health education campaigns should therefore address both primary and secondary symptoms to facilitate timely diagnosis and treatment [71]. Gaps in treatment knowledge also remain a barrier. While recognition of artemether-lumefantrine (AL) as an effective treatment was high in Tanzania (79.7%), it was considerably lower in Kenya (24.3%) [31,45]. Despite chloroquine no longer being recommended as first-line treatment due to widespread resistance in some countries, it continues to be perceived as effective in some communities in India and Nigeria [32,43], illustrating the persistence of inaccurate information.

Community knowledge plays a pivotal role in the uptake of preventive interventions, including ITNs and IRS. While awareness of mosquito nets was high across various settings, actual usage rates varied, often influenced by misconceptions regarding discomfort, side effects, or perceived ineffectiveness [72]. Knowledge of IRS also showed substantial variation, with knowledge exceeding 78.0% in Ethiopia but dropping to 2.7% in Sierra Leone and 3.0% in Kenya [40,44,45]. These disparities reflect inconsistent communication strategies and varying levels of community engagement. Evidence from Mozambique suggests that involving local leaders in promoting IRS can improve acceptance and coverage, demonstrating the value of culturally adapted messaging [51].

The success of other vector control measures, such as larviciding and environmental management, has also been closely linked to active community involvement. In Tanzania and Rwanda, sustained community engagement has supported the adoption of larval source management practices, highlighting the importance of local ownership in implementation [56,57]. Tailoring interventions to cultural contexts enhances their relevance, acceptability, and sustainability of malaria control efforts. Studies indicate that engaging traditional leaders and healers fosters trust and promotes the adoption of preventive practices [5,38]. Similarly, culturally resonant communication methods, such as community performances and arts in Cambodia and folk songs in India, have proven effective in enhancing understanding and retention of malaria messages [73,74]. These examples underscore the importance of incorporating local cultural frameworks into malaria intervention design.

Within-country disparities in malaria knowledge and access to prevention measures require greater attention. Structural barriers in rural and underserved areas continue to limit access to accurate information and essential services. Urban households are generally better informed about malaria, benefitting from proximity to health facilities and increased exposure to mass media platforms such as radio, television, and public health campaigns [52]. In countries such as Equatorial Guinea and Uganda, urban populations exhibit higher knowledge of IRS and mosquito net use compared to rural communities [23,52]. Conversely, rural communities in Malaysia exhibit distinctive malaria knowledge shaped by direct exposure and personal experience with the disease [8]. Furthermore, individuals residing in malaria-endemic regions consistently demonstrate greater malaria knowledge than those in non-endemic areas, likely due to their higher risk and more frequent interactions with control programmes [75].

Sociodemographic factors also influenced community knowledge of malaria. Older individuals tend to be more informed, likely due to their extended exposure to malaria interventions over time [23]. Their accumulated experience with the disease and participation in awareness activities contribute to increased understanding [76]. Educational attainment is another key determinant. Individuals with formal education consistently display a more comprehensive understanding of malaria than those with limited or no education [77]. Prior studies have shown that higher levels of education within communities are associated with improved malaria knowledge [78,79].

Socioeconomic status (SES) is a key determinant of both community knowledge and the capacity to implement malaria prevention practices. Individuals from lower SES backgrounds often have limited access to health information and education. Households with lower incomes or unstable livelihoods frequently face substantial challenges in accessing healthcare services, affording preventive tools such as mosquito nets, and adhering to prescribed treatment regimens [33,65,68]. For example, in Malawi, logistical and financial constraints have hindered the conversion of high awareness into effective preventive behaviours [80]. These findings highlight the need for malaria control programmes to address broader structural

determinants beyond knowledge dissemination. While enhancing community knowledge remains crucial, it must be integrated with efforts to strengthen responsive and resilient health systems that can translate knowledge into action [81]. The long-term success of malaria prevention and treatment hinges on reliable supply chains, fair access to healthcare, and the availability of well-trained and motivated health workers. Behaviour change strategies must therefore be integrated into a health systems framework that supports the sustainability and effectiveness of control interventions [82,83].

This review contributes to the broader literature by offering a global comparison of malaria-related knowledge and behavioural patterns over time and geography. It identifies persistent knowledge gaps and behavioural determinants relevant to malaria control. To translate these findings into effective policy and practice, malaria education should be incorporated into national school curricula in endemic regions. Participatory approaches such as workshops, community seminars, and experiential learning can support deeper understanding and sustained behavioural change [84]. Furthermore, mobile health tools, including SMS alerts, radio broadcasts, and mobile applications, offer scalable platforms for reaching remote or indigenous communities. Community health workers, equipped with linguistically and culturally appropriate tools, can serve as trusted intermediaries between the health system and local communities [85].

## Limitations

This scoping review has several limitations. As it does not critically assess the quality of included studies, potential methodological weaknesses may affect the reliability of the findings. The diversity in study designs and geographical contexts also limits the generalisability of the results, making direct comparisons across regions challenging. Additionally, reliance on peer-reviewed literature introduces publication bias, potentially excluding valuable data from book chapters, posters, and conference abstracts. The restriction to English-language publications further narrows the scope, possibly omitting critical perspectives from non-English-speaking malaria-endemic regions, particularly in Francophone sub-Saharan Africa [86]. Moreover, this review primarily examines short-term knowledge and behavioural responses, without evaluating long-term adherence to malaria prevention strategies. This limits our understanding of the durability of community knowledge and the factors influencing ongoing engagement with control measures.

Another important limitation concerns the lack of disaggregated data. Most studies did not stratify findings by urban–rural location, age group, gender, or sociocultural background, constraining the capacity for subgroup analyses and limiting insights into context-specific knowledge disparities. Similarly, socioeconomic indicators were inconsistently reported, precluding a comprehensive assessment of the role of economic inequality in shaping malaria knowledge and behaviours. Future research should prioritise the systematic collection and reporting of disaggregated data to enhance the equity and contextual relevance of malaria interventions.

In addition, the majority of studies reviewed assessed general community knowledge of malaria symptoms, transmission, and commonly used treatment regimens, without distinguishing among *Plasmodium* species. This reflects a broader gap in the literature, where technical aspects such as species-specific differences in treatment and relapse risk, particularly relevant in regions where both *P. falciparum* and *P. vivax* are co-endemic, are often overlooked. Addressing this gap through public education and targeted research will be essential for advancing species-specific malaria control strategies.

In this review, inter-reviewer agreement during the pilot screening phase was assessed using percent agreement, a measure that does not account for chance agreement. This method was selected to support the training process and ensure consistent application of the inclusion and exclusion criteria, with a pre-defined threshold of over 90% agreement indicating readiness for independent screening. However, we acknowledge the methodological limitation of relying solely on percent agreement and recommend the use of more robust measures of inter-rater reliability, such as Cohen's Kappa, in future reviews.

Despite these limitations, this review provides a comprehensive synthesis of the role of community knowledge in malaria prevention and control programmes. It identifies key knowledge gaps and highlights opportunities for enhancing

community-driven interventions, offering valuable insights to inform future research, policy development, and programme implementation.

## Conclusion

This scoping review underscores the pivotal role of community knowledge in malaria prevention, control, and treatment-seeking behaviour. However, awareness alone is insufficient, bridging the gap between knowledge and action requires a multi-dimensional approach that integrates education, behavioural science, and sustained community engagement. By leveraging local knowledge systems, addressing misinformation, and enhancing access to interventions, malaria control efforts can achieve greater impact and long-term sustainability. Aligning malaria strategies with culturally tailored approaches will empower communities, strengthen health systems, and accelerate progress toward global malaria elimination. A coordinated, community-driven approach, informed by behavioural insights and evidence-based strategies, is essential to transforming malaria from a persistent public health challenge into a controllable and ultimately eliminable disease. Strengthening community participation, fostering cross-sectoral collaboration, and integrating technological advancements will be key to achieving sustainable malaria control and elimination.

## Supporting information

**S1 Checklist.  Preferred Reporting Items for Systematic Reviews and Meta-Analyses extension for Scoping Reviews (PRISMA-ScR) checklist.**
(DOCX)

## Acknowledgments

The authors thank the Director-General of Health, Malaysia for his permission to publish this scoping review.

## Author contributions

**Conceptualization:** Mohd Amierul Fikri Mahmud, Mohd Hatta Abdul Mutalip, Ahmad Mohiddin Mohd Ngesom.

**Investigation:** Faizul Akmal Abdul Rahim, Mohd Amierul Fikri Mahmud, Mohd Hatta Abdul Mutalip, Norzawati Yoep, Mohd Amiru Hariz Aminuddin.

**Methodology:** Faizul Akmal Abdul Rahim, Mohd Hatta Abdul Mutalip, Ahmad Mohiddin Mohd Ngesom.

**Project administration:** Faizul Akmal Abdul Rahim.

**Supervision:** Faizul Akmal Abdul Rahim, Mohd Hatta Abdul Mutalip.

**Writing – original draft:** Faizul Akmal Abdul Rahim, Mohd Amierul Fikri Mahmud.

**Writing – review & editing:** Faizul Akmal Abdul Rahim, Mohd Amierul Fikri Mahmud, Mohd Hatta Abdul Mutalip, Norzawati Yoep, Mohd Amiru Hariz Aminuddin.

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
