## [Decision Letter · Decision Letter 0]

PONE-D-25-20682A Scoping Review on the Impact of Community Knowledge in Malaria ProgrammesPLOS ONE

Dear Dr. Abdul Rahim,

Thank you for submitting your manuscript to PLOS ONE. After careful consideration, we feel that it has merit but does not fully meet PLOS ONE’s publication criteria as it currently stands. Therefore, we invite you to submit a revised version of the manuscript that addresses the points raised during the review process.

We look forward to receiving your revised manuscript.

Kind regards,

Myat Htut Nyunt, MMedSc, PhD

Academic Editor

PLOS ONE

**Journal Requirements:**

1. When submitting your revision, we need you to address these additional requirements. Please ensure that your manuscript meets PLOS ONE's style requirements, including those for file naming. The PLOS ONE style templates can be found at https://journals.plos.org/plosone/s/file?id=wjVg/PLOSOne_formatting_sample_main_body.pdf and https://journals.plos.org/plosone/s/file?id=ba62/PLOSOne_formatting_sample_title_authors_affiliations.pdf 2. We note that your Data Availability Statement is currently as follows: All relevant data are within the manuscript and its Supporting Information files. Please confirm at this time whether or not your submission contains all raw data required to replicate the results of your study. Authors must share the “minimal data set” for their submission. PLOS defines the minimal data set to consist of the data required to replicate all study findings reported in the article, as well as related metadata and methods (https://journals.plos.org/plosone/s/data-availability#loc-minimal-data-set-definition). For example, authors should submit the following data: - The values behind the means, standard deviations and other measures reported;- The values used to build graphs;- The points extracted from images for analysis. Authors do not need to submit their entire data set if only a portion of the data was used in the reported study. If your submission does not contain these data, please either upload them as Supporting Information files or deposit them to a stable, public repository and provide us with the relevant URLs, DOIs, or accession numbers. For a list of recommended repositories, please see https://journals.plos.org/plosone/s/recommended-repositories. If there are ethical or legal restrictions on sharing a de-identified data set, please explain them in detail (e.g., data contain potentially sensitive information, data are owned by a third-party organization, etc.) and who has imposed them (e.g., an ethics committee). Please also provide contact information for a data access committee, ethics committee, or other institutional body to which data requests may be sent. If data are owned by a third party, please indicate how others may request data access. 3. When completing the data availability statement of the submission form, you indicated that you will make your data available on acceptance. We strongly recommend all authors decide on a data sharing plan before acceptance, as the process can be lengthy and hold up publication timelines. Please note that, though access restrictions are acceptable now, your entire data will need to be made freely accessible if your manuscript is accepted for publication. This policy applies to all data except where public deposition would breach compliance with the protocol approved by your research ethics board. If you are unable to adhere to our open data policy, please kindly revise your statement to explain your reasoning and we will seek the editor's input on an exemption. Please be assured that, once you have provided your new statement, the assessment of your exemption will not hold up the peer review process. 4. Your ethics statement should only appear in the Methods section of your manuscript. If your ethics statement is written in any section besides the Methods, please move it to the Methods section and delete it from any other section. Please ensure that your ethics statement is included in your manuscript, as the ethics statement entered into the online submission form will not be published alongside your manuscript.

**Additional Editor Comments:**

The study does not directly assess the impact of community knowledge on malaria outcomes. Instead, it maps and summarizes the existing literature on community knowledge regarding malaria prevention and control. Thus, "impact" should be omitted in the title.

Additionally, the abstract and introduction suggest the review aims to assess the impact of community knowledge, but the methods only describe mapping knowledge, not evaluating outcomes or effectiveness. In the methods, clarify that the review did not extract or analyze data on the impact of knowledge, but rather on the extent and nature of community knowledge. Clarify it.

Reviewers' comments:

Reviewer's Responses to Questions

**Comments to the Author**

1. Is the manuscript technically sound, and do the data support the conclusions?

Reviewer #1: Partly

Reviewer #2: Yes

2. Has the statistical analysis been performed appropriately and rigorously? 

Reviewer #1: N/A

Reviewer #2: Yes

3. Have the authors made all data underlying the findings in their manuscript fully available?

Reviewer #1: No

Reviewer #2: Yes

4. Is the manuscript presented in an intelligible fashion and written in standard English?

Reviewer #1: Yes

Reviewer #2: Yes

5. Review Comments to the Author

**Reviewer #1:**  Title

(1) For the title, instead of “Malaria Programmes,” change to “Malaria Prevention” or “Malaria Prevention and Control Outcomes” or “Malaria Control and Elimination Efforts” or the term being more specific.

Abstract

(1) The abstract has a clear structured format: Background, Methods, Results, Conclusion.

(2) For methods, briefly describe the inclusion criteria (e.g., only peer-reviewed? specific populations?).

(3) For results, briefly mention examples for “critical knowledge gaps”

Introduction

(1) The introduction is well-structured and informative. However, transitions between some paragraphs are abrupt (e.g., from biomedical approaches to socioeconomic status). Use linking sentences to maintain a smooth flow.

Material and Methods

(1) Studies were included for original research published in English and exclusion of non-English studies introduces language bias, especially for malaria-endemic regions and excluding conference proceedings etc. may omit emerging evidence, which can be critical in rapidly evolving health issues like malaria. The community-based nature of the topic, reports from NGOs, ministries of health, or unpublished evaluations could provide valuable insights.

(2) The manuscript mentioned that “a pilot sample of 20 articles was used to train the screeners, with a concordance rate exceeding 90% indicating their readiness to proceed.” Describe how agreement was measured (e.g., Cohen’s Kappa).

(3) Only three databases (PubMed, Scopus and Web of Science) were searched. Justify why only these databases were used or expand the search to include additional sources to ensure a comprehensive literature review.

Results

(1) The results rely heavily on text and tables. Add summary tables or figures to present key findings more effectively for country-specific knowledge levels and practices. Include a summary of how knowledge or prevention behavior has changed over time.

(2) In Table (1), the year of the authors are randomly described. It is better to describe chronologically for clarity.

(3) “Community knowledge of malaria transmission varied across regions, with 54 studies,” “Community knowledge of malaria symptoms was assessed in 40 studies,” “Community knowledge of malaria prevention and control measures was assessed in 53 studies,” regarding with these, why varied with 40, 53, 54 studies etc. instead of 62 studies? Describe the reasons for studies not include all studies areas.

(4) “General awareness of antimalarial drugs was noted by 73.3% of respondents in Nigeria [47], 81.6% in Ethiopia [30], and 90% in Malaysia [8], reflecting varying levels of accessibility and health education initiatives.” In relation to these questions, were respondents aware of any species differences, such as Plasmodium falciparum and Plasmodium vivax, in terms of antimalarial drug treatment (Pf for AL and Pv for CQ)?

Discussion

(1) Some statements, like “Community knowledge levels strongly influenced the ownership and use of mosquito nets...” assume broad regional trends without specifying the variability within countries or among different community types (urban vs. rural, indigenous vs. non-indigenous, etc.). If possible, quantify or categorize heterogeneity (e.g., rural vs urban settings, year ranges) the findings and discuss or acknowledge it.

(2) The discussion does not sufficiently link the findings to practical applications or broader policy implications. Compare the findings more explicitly to prior reviews or studies, particularly those reporting conflicting or supporting evidence, and explain how this review contributes uniquely to the field. Discuss more targeted suggestions, such as: incorporating malaria education into school curricula, utilizing mobile health tools to reach remote areas and training local community health workers to sustain education efforts.

(3) The discussion does not mention how gender, age, or educational level affect community knowledge or malaria prevention practices. These demographic factors are known to significantly influence health behaviour and access to information, discuss it, if data are available.

(4) The discussion focuses mainly on community knowledge, but less so on the health system’s role for behaviour change. Although community education is vital, it must be supported by a strong health system including reliable supply chains, well-trained personnel, and accessible facilities to ensure that increased awareness leads to translate knowledge into effective action. Add those in discussion.

(5) There's little discussion of global comparisons that how the findings from Africa and Asia compare to or inform malaria responses in other parts of the world.

**Reviewer #2:**  The manuscript presents a systematic review exploring the impact of community knowledge on malaria control across 27 countries. I have some minor comments and attached my comments and suggestions in an attached PDF file.

6. PLOS authors have the option to publish the peer review history of their article (what does this mean? ). If published, this will include your full peer review and any attached files.

**Do you want your identity to be public for this peer review?** For information about this choice, including consent withdrawal, please see our Privacy Policy .

Reviewer #1: No

Reviewer #2: No

---

## [Author Response · Author response to Decision Letter 1]

19 Jun 2025

We have provided our responses to the reviewers’ comments in the ‘Response to Reviewer’ file, which is included in the attachment section. Thank you.

---

## [Decision Letter · Decision Letter 1]

PONE-D-25-20682R1A Scoping Review of Community Knowledge in Malaria Prevention and Control ProgrammesPLOS ONE

Dear Dr. Abdul Rahim,

Thank you for submitting your manuscript to PLOS ONE. After careful consideration, we feel that it has merit but does not fully meet PLOS ONE’s publication criteria as it currently stands. Therefore, we invite you to submit a revised version of the manuscript that addresses the points raised during the review process.

Please revise minor points according to reviewer 1's comment and resubmit it.

We look forward to receiving your revised manuscript.

Kind regards,

Myat Htut Nyunt, MMedSc, PhD

Academic Editor

PLOS ONE

Journal Requirements:

Additional Editor Comments:

Please revise minor points according to the Reviewer 1 comment.

Reviewers' comments:

Reviewer's Responses to Questions

**Comments to the Author**

1. If the authors have adequately addressed your comments raised in a previous round of review and you feel that this manuscript is now acceptable for publication, you may indicate that here to bypass the “Comments to the Author” section, enter your conflict of interest statement in the “Confidential to Editor” section, and submit your "Accept" recommendation.

Reviewer #1: (No Response)

Reviewer #2: All comments have been addressed

2. Is the manuscript technically sound, and do the data support the conclusions?

Reviewer #1: Yes

Reviewer #2: Yes

3. Has the statistical analysis been performed appropriately and rigorously? 

Reviewer #1: N/A

Reviewer #2: Yes

4. Have the authors made all data underlying the findings in their manuscript fully available?

Reviewer #1: Yes

Reviewer #2: Yes

5. Is the manuscript presented in an intelligible fashion and written in standard English?

Reviewer #1: Yes

Reviewer #2: Yes

6. Review Comments to the Author

Reviewer #1: Review comments

Title

The authors have revised the title as suggested.

Abstract

The authors have revised as suggested.

Introduction

The authors have revised as suggested. However, the transition sentence beginning with “Moreover, it is increasingly …” to introduce socioeconomic status after discussing biomedical approaches remains abrupt. Instead of using “Moreover”, it is better to change as “Understanding these non-biomedical influences, it is important to consider the broader social and economic factors that influence health behaviors and access to healthcare services.”

Material and Methods

(1) Regarding with Q1, the authors have revised as suggested.

(2) For Q2, add a sentence specifying the method of agreement measurement or alternatively, if this was not done, they should acknowledge the limitation and it will consider more measures in future reviews.

(3) For Q3, add a sentence for rationale of why choosing only three databases such as their broad coverage of peer-reviewed biomedical literature and clarify whether additional sources were considered to ensure a comprehensive review.

Results

(1) Regarding with Q1, the authors added a summary Table 2.

(2) Regarding with Table (1), the authors have revised as suggested. However, the formatting still needs to be standardized, particularly the alignment and line spacing in the columns titled “Community Knowledge of Malaria” and “The Role of Community Knowledge.”

(3) Regarding with Q3, the authors should clarify in the main text why not all studies assessed every aspect of community knowledge. For example, they could include a statement such as: “Although our review included 63 studies, not all of them assessed every aspect of community knowledge. Specifically, knowledge of malaria transmission was evaluated in 54 studies, symptoms in 40 studies, and prevention or control measures in 53 studies, depending on each study’s focus and objectives.”

(4) For Q4, the authors have revised the issue by adding it to the limitations section.

Discussion

(1) For Q1, the authors have revised as suggested. It is satisfactory.

(2) For Q2, the authors have revised as suggested.

(3) The authors have revised as suggested for Q3.

(4) The authors have added the discussion as suggested for Q4.

(5) For Q5, the authors have revised as suggested.

Reviewer #2: (No Response)

7. PLOS authors have the option to publish the peer review history of their article (what does this mean? ). If published, this will include your full peer review and any attached files.

**Do you want your identity to be public for this peer review?** For information about this choice, including consent withdrawal, please see our Privacy Policy .

Reviewer #1: No

Reviewer #2: No

---

## [Editor Report · Decision Letter 2]

A Scoping Review of Community Knowledge in Malaria Prevention and Control Programmes

PONE-D-25-20682R2

Dear Dr. Abdul Rahim,

We’re pleased to inform you that your manuscript has been judged scientifically suitable for publication and will be formally accepted for publication once it meets all outstanding technical requirements.

Kind regards,

Myat Htut Nyunt, MMedSc, PhD

Academic Editor

PLOS ONE
---

## [Editor Report · Acceptance letter]

PONE-D-25-20682R2

PLOS ONE

Dear Dr. Abdul Rahim,

I'm pleased to inform you that your manuscript has been deemed suitable for publication in PLOS ONE. Congratulations! Your manuscript is now being handed over to our production team.

Kind regards,

on behalf of

Dr. Myat Htut Nyunt

Academic Editor

PLOS ONE